# Recent Advances in Bacterial Degradation of Hydrocarbons

**Emiliana Pandolfo, Anna Barra Caracciolo *** and **Ludovica Rolando**

Water Research Institute, National Research Council (IRSA-CNR), 00010 Rome, Italy
* Correspondence: barracaracciolo@irsa.cnr.it

**Abstract:** Hydrocarbons occur in fossil fuels such as crude oil and consist mainly of hydrogen and carbon. Although they are natural chemicals, crude oil refining results in commercial products with new physico-chemical properties, which can increase their complexity and toxicity, and hamper their degradation. The presence of biodiverse natural microbial communities is a prerequisite for an effective homeostatic response to the various hydrocarbons, that contaminate ecosystems. However, their removal depends on the compartment contaminated (water, sediment, soil), their molecular weight, and their toxicity not hampering microbial activity. This paper reports different bacterial species involved in the biodegradation of aliphatic and aromatic hydrocarbons. Hydrocarbon contamination is generally due to the co-presence of a mixture of these chemicals, and their removal from the environment cannot rely on only a single species but generally requires bacterial consortia. Versatile bacterial metabolism relies on specific genes encoding the key enzymes involved in the peripheral metabolic and central metabolic pathways for degrading aliphatic and polycyclic aromatic hydrocarbons. Although microbial metabolism can have the potential for natural attenuation of these contaminants, hydrocarbon bioremediation, through biostimulation (e.g., use of surfactants, plants, earthworms, and nanoparticles) and bioaugmentation, can be a valid tool for removing them from actually contaminated soil, freshwater, groundwater, and seawater.

**Keywords:** microbial communities; regulation ecosystem services; aliphatic hydrocarbons; polycyclic aromatics hydrocarbons; peripheral metabolic pathway; central metabolic pathway; bioaugmentation; biostimulation

## 1. Introduction

Economic activities are strongly dependent on the use of fossil fuels, which consist of hydrocarbon-containing material formed naturally in the Earth's crust from dead plant and animal residues. Hydrocarbons provide energy for civil and industrial purposes, and for transport [1]. Owing to the growth of the human population, the worldwide demand for hydrocarbons is increasing, and petrochemical industries, accidental oil spills, disconnection between oil wells and combustion processes (e.g., industry emissions), abandoned refining sites, and vehicle combustion have been constantly polluting air, water, and soil [2–4]. The risk of accidental leaks in ecosystems has increased exponentially with the growing global demand for oil and it has been estimated that every year, ca. 1.3 million liters of oil reach natural environments [5]. Hydrocarbons are the organic contaminants most commonly found in ecosystems [1,6], and it is fundamental to evaluate their environmental fate and effects in different matrices.

Hydrocarbons are chemicals naturally occurring in crude oil and consisting mainly of hydrogen and carbon. Crude oil is a dark, viscous, and easily flammable complex liquid mixture of hydrocarbons (83–87%), comprising variable amounts of hydrogen (10–14%), oxygen (0.05–1.5%), sulfur (0.005–6.0%), nitrogen (0.1–0.2%), and metals (<1000 mg/L) such as nickel, iron, and copper. The specific composition depends on the oil field's geological age, location, and depth. After crude oil refining, the resulting products have new physico-chemical properties, which increase their complexity and can hamper their biodegradation [7].

Microorganisms play key roles in natural ecosystem functioning, such as primary production, organic matter decomposition, nutrient cycling, and biodegradation of contaminants, including hydrocarbons, thus contributing in different ways to soil and water purification processes. The maintenance of these regulating ecosystem services is linked to bacterial diversity and metabolic versatility, which makes it potentially possible to biodegrade a huge variety of aliphatic and aromatic hydrocarbons.

## 2. Aliphatic Hydrocarbons and Polycyclic Aromatic Hydrocarbons

Hydrocarbons can be divided in accordance with their chemical structure into: aliphatic hydrocarbons or saturated hydrocarbons, aromatic hydrocarbons (monocyclic aromatic and polycyclic aromatic hydrocarbons); and heteroatomic compounds (saturated and aromatic ones), including resins and asphaltenes.

Aliphatic hydrocarbons (AH) (Figure 1A), also known as paraffins, are mainly present in deposits of natural gas and oil formed by plant and animal decomposition. AH have no double bonds (general formula $C_nH_{2n}$) and represent the highest percentage of the constituents of crude oil. The lack of functional groups makes them strongly apolar, so they have low water solubility and are poorly reactive at room temperature [8]. In accordance with their structure, they are classified as alkanes (which have a linear structure) and cycloalkanes (which have a condensed ring structure); they can be linear or branched [9].

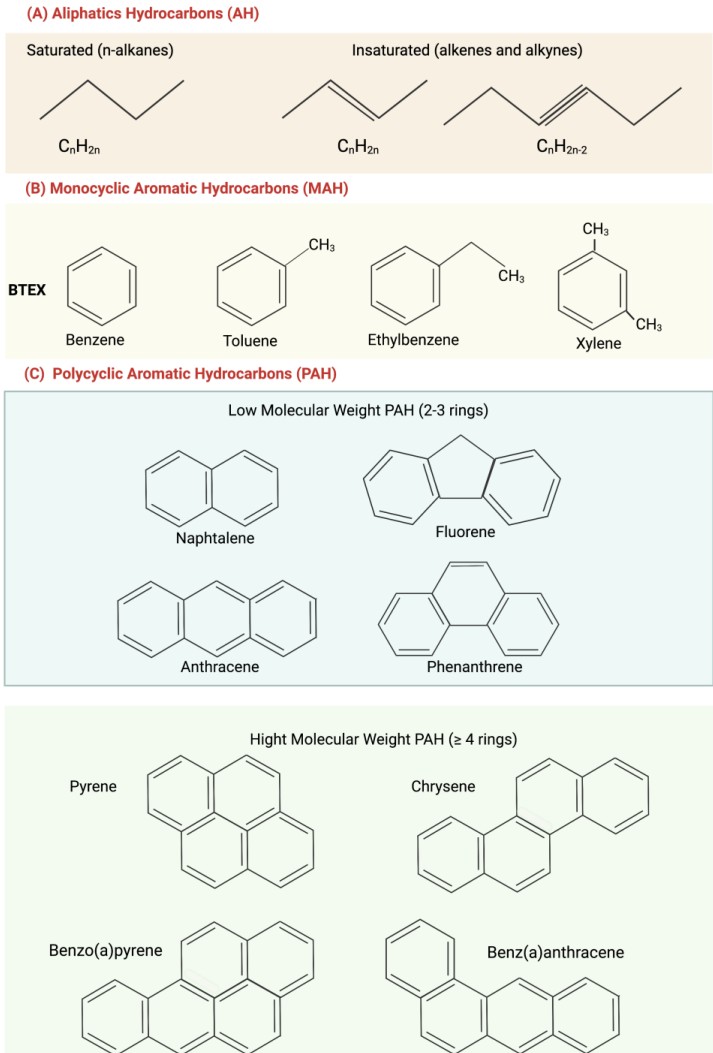

**Figure 1.** Structure of main hydrocarbons: (**A**–**C**), subdivided into low molecular weight (in blue) and high molecular weight (in green).

Aromatic hydrocarbons have one or more benzene rings and are generally replaced with different aliphatic hydrocarbons. They are mainly divided into monocyclic aromatic hydrocarbons (MAH), such as benzene, toluene, ethylbenzene, and xylene, which together constitute BTEX (widely studied and making up 2–20% of oil) (Figure 1B) and polycyclic aromatic hydrocarbons (PAH), (Figure 1C).

PAHs are a group of lipophilic organic pollutants that derive from biological processes or are formed as products of incomplete combustion from natural (forest fires and shrubs) or anthropic (vehicle emissions, domestic heating, and cigarette smoke) sources. Owing to their ubiquitous presence and persistence in air, water, and soil, as well as their toxicity for both humans and biota, they are compounds of environmental concern [10,11].

PAHs with 2–3 benzene rings, such as naphthalene, fluorene, anthracene, and phenanthrene, are low-molecular-weight ones. PAHs with four or more benzene rings, such as pyrene, chrysene, benzo(a)pyrene, and benz(a)anthracene, are high molecular weight ones (Figure 1C). Low-molecular-weight PAHs are gases and tend to escape into the atmosphere, while those with a higher molecular weight are liquids or in a solid state at room temperature and often have a separate phase in water [8]. PAH solubility in aqueous solution decreases as the number of benzene rings increases, and this increases their environmental persistence. A high persistence increases the possibility of a compound making its toxicity felt.

It has recently been recognized that life quality is connected to that of the environment, as expressed by the new concept of 'One Health' proposed by the World Health Organization [12]. The One Health approach means that human health is connected to that of animals and the environment. For this reason, the ubiquitous occurrence (water, air, soil, and sediment) of hydrocarbons is a serious threat to both human and environmental health [13].

## 3. Environmental Fate and Toxic Effects

Hydrocarbons interact with both the abiotic and biotic components of ecosystems. Hydrocarbons can be divided into specific fractions, equivalent to the number of carbon atoms. These fractions can also be described for their physical, chemical, and toxicological characteristics. Fraction 1 (F1) contains hydrocarbons with a carbon number between 6 and 10 (C6–C10) and is classified as volatiles. Fraction 2 (F2) represents the semi-volatile hydrocarbons with at least 10 carbon atoms (C10–C16). Fraction 3 (F3) includes hydrocarbons with a carbon number between 16 and 34 (C16–C34) and are considered non-volatile. Fraction 4 (F4) refers to the hydrocarbon class with the lowest volatility and solubility, and more than 35 carbon atoms (>C35). Toxicity generally increases with the molecular weight [7,9]. For example, polycyclic aromatic hydrocarbons with a high molecular weight and boiling point have higher toxicities [7].

Due to their complex characteristics, the lightest and most volatile hydrocarbon fractions are released into the atmosphere, the amphipathic and hydrophilic fractions dissolve in water, and the lipophilic ones tend to bind to soil/sediment particles and organic matter [7].

The toxicity of a hydrocarbon also depends on its bioavailability, which in turn is influenced by its physical and chemical characteristics. Once a bioavailable chemical is absorbed by an organism, it interacts with cellular receptor sites and can cause lethal or sub-lethal effects. The effects depend on the concentration and duration (short or long) of exposure [7]. Lethality occurs from short-term exposure due to acute oil spills in aquatic environments in which hydrophobic compounds destroy the central nervous systems of organisms through their location in neuron cell membranes [7].

Sub-lethal toxic effects due to hydrocarbon exposure can cause lesions (internal or external), defects in early development, anoxia, and changes in molecular and behavioral functions related to nutrition and reproduction. Exposure to petroleum hydrocarbon products can cause acute effects such as changes in the nervous, ocular, and respiratory systems. Along with migraines and headaches, people living in oil-polluted areas also experience nausea, upper respiratory tract infections, and nose and eye irritations [5].

PAHs can potentially not only affect the nervous, immune, and excretory systems but also cause tumors and mutations [7,9]. Chronic exposure to PAHs can cause cancer in humans, and the cataractogenic properties of PAHs (i.e., alterations at the dermal and ocular levels) have been well documented [14]. Exposure to PAHs can occur through inhalation, ingestion (through contaminated food or drink), or dermally (direct contact). Once introduced into the body, these molecules rapidly diffuse due to their high fat solubility, which makes them able to cross cell membranes and deposit themselves in adipose tissue, the kidneys, and the liver. Exposure to these substances causes blood damage, and probable immunosuppression. Depending on the route of exposure, PAHs (particularly those characterized by 3–7 aromatic rings) cause lung, digestive, or skin cancers. The carcinogenic potential derives from the metabolism of these substances in the liver, which leads to a biotransformation into reactive intermediates capable of binding to DNA and RNA with consequent damage to genetic information. It has been empirically demonstrated that PAHs interact metabolically, resulting in synergistic and additive effects [15].

The toxicity of hydrocarbons affects humans, plants, animals, and microorganisms, with consequences for ecosystem biodiversity and functioning. Environmental contamination by hydrocarbons and derivatives has caused the local extinction of many species of plants and animals [5].

Hydrocarbons are freshwater, wastewater, and seawater contaminants. Because they contain a large number of hydrophobic components, which are adsorbed onto particulates and sediments in aquatic ecosystems, they can become bioavailable for benthic species. In fact, invertebrates, fish, eggs laid by fish, and filter feeders such as bivalves are in contact with these contaminants by filtration of suspended petroleum [7].

Hydrocarbons can hamper water and oxygen transfer through porous spaces in sediments and soil, influencing their permeability, moisture content, pH, nutrient availability (e.g., nitrate, phosphate, and sodium), and redox condition [3]. This is the case of higher molecular weight PAHs, which thanks to their low desorption from soil and low water solubility can form a surface layer that can prevent both bioaccessibility and vegetation development for several decades [16]. Damaging effects also occur on plant species directly exposed to hydrocarbons because hydrocarbons prevent light access and make plants unable to acquire nutrients and water, reducing primary and agricultural productivity [5,17].

Hydrocarbons can also be released into deeper waters from soil by percolation or atmospheric subsidence, bringing them into contact with underground ecosystems [18].

Hydrocarbons can also eliminate or inhibit microbial species, altering their corresponding ecosystem functions [17]. The high selective pressure exerted by hydrocarbons can reduce their diversity, producing rapid shifts in microbial populations [19]. The elimination or alteration of microbial species with key roles in biogeochemical cycles and primary production can also affect higher trophic levels of the food chain due to biomagnification [17]. Bacteria in soil, e.g., nitrifying bacteria, are susceptible to hydrocarbon exposure, which inhibits their enzyme ammonium monooxygenase, through competitive binding with low-weight hydrocarbons [7], with consequences for soil fertility.

Several hydrocarbons have been classified as priority and hazardous substances. In fact, eight PAHs (anthracene, benzo(a)pyrene, benzo(b)fluoranthene, benzo(g,h,i)perylene, benzo(k)fluoranthene, fluoranthene, indeno(1,2,3-cd)pyrene and naphthalene) have been included in the priority and hazardous substances list of the WFD (2000/60/EC). Moreover, the 2013/39/EU Directive stated that benzo(a)pyrene can be considered an overall PAH marker, and, based on its toxicity values, also established environmental quality standards (EQS) for biota and water (AA-EQS) [20], (Table 1).

Hydrocarbons act as a selective force, influencing the structure and functioning of natural microbial communities. Some microbial species are negatively affected by these toxic compounds [7]. However, others can respond to their presence with various mechanisms and remove them. Hydrocarbon biodegradation is possible if a high microbial metabolic

diversity is present in a contaminated environment and the quantity of hydrocarbons is not so great as to prevent or inhibit the activity of the microbes [17].

**Table 1.** Environmental quality standards (EQS) for PAHs. AA = annual average, MAC = maximum average concentration (2013/39/EU Directive).

| Substance Name | CAS Number | AA-EQS Surface Waters (µg/L) | MAC-EQS Inland Surface Waters (µg/L) | MAC-EQS Other Surface Waters (µg/L) | EQS Biota (µg/kg) |
|---|---|---|---|---|---|
| Anthracene | 120-12-7 | 0.1 | 0.1 | 0.1 | |
| benzo(a)pyrene | 50-32-8 | $1.7 \times 10^{-4}$ | 0.27 | 0.027 | 5 |
| benzo(b)fluoranthene | 205-99-2 | * | 0.017 | 0.017 | * |
| benzo(g,h,i)perylene | 207-08-9 | * | $8.2 \times 10^{-3}$ | $8.2 \times 10^{-4}$ | * |
| benzo(k)fluoranthene | 191-24-2 | * | 0.017 | 0.017 | * |
| Fluoranthene | 206-44-0 | 0.0063 | 0.12 | 0.12 | 30 |
| indeno(1,2,3-cd)pyrene | 193-39-5 | * | - | - | * |
| Naphthalene | 91-20-3 | 2 | 130 | 130 | |

Note: * EQS biota and AA-EQS water values refer to the concentration for benzo(a)pyrene.

## 4. Microbial Degradation of Hydrocarbons

Methane, the simplest hydrocarbon, can be degraded by a highly specialized group of bacteria, the methanotrophic one, which uses it as a carbon and energy source. However, these bacteria are unable to grow on hydrocarbons containing more carbon atoms [21]. The complex high-molecular-weight pyrene is highly hydrophobic, has low water solubility, and tends to sorb to soil organic carbon; these characteristics give it low bioavailability and make it recalcitrant to biodegradation [22]. Pyrene degradation was reported first in 1988, involving a *Mycobacterium* strain capable of mineralizing and using it as its sole carbon and energy source [23]. Subsequently, several pyrene degraders have also been identified, belonging to the genera *Sphingomonas*, *Mycobacterium*, *Rhodococcus*, *Bacillus*, *Burkholderia*, *Cycloclasticus*, *Pseudomonas*, and *Stenotrophomonas*, with degradation percentages in soil ranging from 32% to 96% with variable times (4–42 days) [22,24].

If large quantities of oil (hydrocarbon mixture) are released into the environment, the volatile hydrocarbon fraction evaporates rapidly, and longer aliphatic chains and aromatic compounds remain. Because oil is insoluble and less dense than water, it floats on the surface and forms blotches. Hydrocarbon-oxidizing bacteria need to adhere to insoluble petroleum droplets, on which a large number of cells can be observed. The activities of these organisms lead to the degradation of oil and the breakdown of oily patches [25].

Numerous studies have shown that several bacterial communities in contact with hydrocarbons rapidly shift to bacterial species able to degrade and utilize hydrocarbon compounds as sources of carbon and energy (hydrocarbonoclastic bacteria) [19]. Hydrocarbonoclastic bacteria have evolved adaptive mechanisms to tolerate the presence of hydrocarbons, such as the ability to emulsify and metabolize them, activation of DNA repairing mechanisms, production of the molecules involved in the mechanisms of quorum sensing and biofilm, and regulation of efflux pumps and pores to control the concentration of hydrocarbons inside a cell [19].

Biodegradation of a hydrocarbon involves adhesion to the substrate and production of compounds such as biosurfactants/bioemulsionants, biopolymers, solvents, gases, and acids for making them bioavailable [9]. In particular, biosurfactants are high- or low-molecular-weight amphiphilic molecules that can be synthesized by numerous microorganisms. In prokaryotes, the ability to produce these compounds is often matched with the ability to grow on insoluble substrates such as hydrocarbons; the production and function of these compounds are in fact closely related to the mechanisms of absorption

(uptake) of such substances inside bacterial cells. These molecules often have the property of detoxifying the substrate [8].

Posada-Baquero et al. [26] showed that the sunflower rhizosphere, in the presence of a biosurfactant (rhamnolipid), favored the desorption of PAHs in a creosote-contaminated soil. Moreover, the enhancement of slow desorption of PAHs resulted in faster biodegradation in a slurry biodegradation experiment.

Chebbi et al. (2017) [27] isolated a *Pseudomonas* strain (W10) from a diesel-contaminated soil through cultures enriched with phenanthrene. This strain showed potential growth in the presence of a broad group of hydrocarbons, including aliphatic, monocyclic aromatic, and polycyclic aromatic hydrocarbons. This was due to the strain's ability to synthesize a biosurfactant, which made hydrocarbons readily available for their degradation. In another study [28], *Pseudomonas aeruginosa* was able to degrade n-alkanes (C16 and C19) and polycyclic aromatic hydrocarbons (PAHs) such as fluorene, phenanthrene, and pyrene.

Other authors [29] isolated *Enterobacter cloacae* strains with a high capacity to emulsify hydrocarbons, lower surface tension, and increase the rate of degradation of these contaminants.

The genus *Delftia* has also been identified for hydrocarbon degradation. For example, *Delftia* sp. NL1, isolated from an oil field in Algeria, degraded more than 66.76% of diesel in 7 days, producing glycolipids that acted as biosurfactants and improving the accessibility and bioavailability of the insoluble hydrocarbon fraction [30].

Another biosurfactant producer, the *Achromobacter* (AC15) bacterium, was isolated and was able to use high concentrations of pyrene as its sole carbon and energy source. It degraded (in 14 days) about 40% of the pyrene supplied at an initial concentration of 300 mg/L because it produced bioemulsionants able to reduce the culture medium surface tension (from 67.2 to 33.2 mN/m) [31].

Bacteria belonging to the genus *Acinetobacter* also have a key role in PAH biodegradation, as reported in a recent study where the increase in their relative abundance was demonstrated in the presence of diesel, heavy metals, and PAHs [32]. The *Acinetobacter* genus is generally known for its efficiency in metabolizing various hydrocarbons such as monoaromatic compounds [33], and polyaromatic compounds such as naphthalene [34], acenaphthene, and acenaphthylene [11,35].

In some bacteria, a hydrocarbon presence can modulate their chemotaxis [7]. Chemotaxis is a behavioral response where bacteria perceive variations in the concentration of a specific chemical using chemoreceptors and respond to them by changing their position. Bacteria able to degrade hydrocarbons can control their position and migrate towards contaminated points [7,14]. For example, the motile and chemotactic bacterium *Pseudomonas putida* G7 has recently been found to sorb and cometabolize pyrene in a column experiment, mobilizing and making it bioavailable for biodegradation [36].

The *Stenotrophomonas* genus is known to be able to use hydrocarbons as the sole carbon and energy source. Elufisan et al. [37] isolated *Stenotrophomonas* sp. *Pemsol* from a soil contaminated with crude hydrocarbons and showed its capability to grow in the presence of five different PAHs (biphenyl, anthraquinone, phenanthrene, naphthalene, and phenanthridine) as the sole source of carbon and energy. The complete genome of this sequenced strain revealed the presence of 145 genes involved in PAH degradation. The same authors also reported that some genes associated with the catalytic metabolism of hydrocarbons can be acquired by other bacteria by horizontal gene transfer.

In addition to *Proteobacteria*, another phylum including several hydrocarbon-degrading strains is *Actinobacteria* [19]. The strain *Rhodococcus* sp. P14 has been reported to be able to degrade high molecular weight PAHs (3, 4, or 5 aromatic ring molecules such as phenanthrene, pyrene, and benzo(a)pyrene), as well as aliphatic hydrocarbons. The overall fatty acid composition of the cell membrane of *Rhodococcus* sp. P14 was altered when the strain was grown in enrichment cultures with different types of hydrocarbons, leading to a general decrease in short-chain fatty acids and an increase in branched-chain fatty acids. This happens to make the surface of the bacterial cells more hydrophobic in order to facilitate non-water-soluble hydrocarbon use. It has also been observed that this

bacterial strain is able to form a biofilm between the surface of an aqueous solution and that of insoluble hydrocarbons [38]. In another study, the degrading potential of the genus *Rhodococcus* was shown. Indeed, *Rhodococcus* sp. WAY2 was found to be specialized in the metabolism of short-chain alkanes [39].

The overall bacterial genera involved in hydrocarbon biodegradation in the studies reported above are summarized in Table 2.

**Table 2.** Example of single bacteria involved in hydrocarbon degradation.

| | Genus/Species | Mechanism of Action/ Hydrocarbon | References |
|---|---|---|---|
| *Proteobacteria* | *Pseudomonas W10* | Biosurfactant production/ phenanthrene | [25] |
| | *Pseudomonas aeruginosa* | Biosurfactant production/ n-alkanes (C16–C19), fluorene, phenanthrene, pyrene | [26] |
| | *Enterobacter cloacae* | Biosurfactant production/ crude oil | [27] |
| | *Delftia* sp. *NL1* | Biosurfactant production/diesel | [28] |
| | *Achromobacter (AC15)* | Biosurfactant production/ Pyrene | [29] |
| | *Acinetobacter* | Biosurfactant production/ naphthalene, acenaphthene, acenaphthylene | [11,32,33] |
| | *Pseudomonas putida* | Chemotaxis/ Pyrene | [36] |
| | *Stenotrophomonas* sp. *Pemsol* | Horizontal gene transfer/ biphenyl, anthraquinone, phenanthrene, naphthalene, phenanthridine | [37] |
| *Actino bacteria* | *Rhodococcus* sp. *P14* | Change in fatty acid composition of cell membrane/biofilm formation/phenanthrene, pyrene, benzo(a)pyrene | [38] |

Due to their complexity and variety, the removal of hydrocarbons from the environment cannot rely on a single species but requires versatile bacterial consortia. Indeed, a metabolite produced by a bacterial strain may be further degraded by another one of the same consortium [9,40]. The final biodegradation products can be converted into inorganic substances, nutrients, and cellular biomass [41]. For example, pyrene can be efficiently degraded by bacterial consortia [42–44] and naphthalene, benzo[a]pyrene have been found to be biodegraded by bacterial inocula, previously enriched on wheat straw, because of the similarity of PAH rings to those occurring in lignocellulosic biomass [45].

Phulpoto et al. [46] isolated bacterial consortia from different depths of a lake (i.e., surface water, deep water, and sediments) and tested their bioremediation capacity in laboratory experiments. The most dominant hydrocarbon-degrading genera were *Pseudomonas*, *Acinetobacter*, and *Stenotrophomonas* in the surface water, *Acinetobacter*, *Aeromonas*, *Sphingobacterium*, and *Pseudomonas* in the sediment, and *Acinetobacter*, *Pseudomonas*, *Comamonas*, *Flavobacterium*, and *Enterobacter* in the deep water (6 m depth). They demonstrated that biodegradation was more efficient using consortia comprising 3–5 different bacterial species. Interestingly, they found the highest degradation percentages in the sediment (67.60% in 12 days) and deep water (59.70% in 12 days) due to the increase in biosurfactant production by the bacterial consortia there.

Several bacterial consortia able to degrade different types of spilled oil have also been identified in seawater. A consortium consisting of *Burkholderia*, *Rhodanobacter*, and *Pseudomonas aeruginosa* was able to efficiently remove diesel (>C12), and another one, comprising mainly *Flavobacterium* sp., *Acinetobacter calcoaceticus*, and *Pseudomonas aeruginosa*, was able to biodegrade engine oil/lubrificating oil (C9–C16) in seawater [47,48].

Bacosa et al. [49] isolated a kerosene-degrading consortium (*Achromobacter*, *Alcaligenes*, and *Cupriavidus* genera) in contaminated seawater. The consortium was able to degrade preferentially aromatic hydrocarbons, although their toxicity is higher than aliphatic ones.

Jamal et al. [50] isolated different *Marinobacter* strains (*Marinobacter hydrocarbonoclasticus* strain MF716467, *Marinobacter* sp. MF716468, and *Marinobacter hydrocarbonoclasticus* strain MF716469) in the Red Sea in Saudi Arabia. The consortium was able to degrade phenantrene (initial concentration: 1500 mg/L) and pyrene (initial concentration: 300 mg/L). The degradation of these was 72% (at 16 days) for phenantrene and 86% (at 12 days) for pyrene, respectively.

Although microbial metabolism has the potential to naturally attenuate contaminated environments and biostimulate hydrocarbons (e.g., using surfactants, plants, or nanoparticles) or bioaugment them, it is more effective in their removal.

For example, bioremediation of a petroleum hydrocarbon-contaminated soil collected from an abandoned plant in China was performed successfully in batch experiments combining an isolated indigenous bacterial consortium (*Enterococcus*, *Vagococcus*, and *Sphingomonas*) and the sophorolipid biosurfactant. The surfactant enhanced bioremediation with increasing concentrations of sophorolipid because it improved hydrocarbon bioaccessibility for microorganisms [51].

Bioremediation of hydrocarbons can also be significantly enhanced by using plants (bioassisted phytoremediation). Vasilyeva et al. [52] showed significant reductions in total petroleum hydrocarbon content in a contaminated crude oil soil (5–15% *w/w*) using a mixed adsorbent (ACD), composed of granular activated carbon and diatomite, in combination with a biopreparation (BP) consisting of hydrocarbon-degrading bacteria (*Pseudomonas putida* B-2187 and *Rhodococcus erytropolis* Ac-859). The ACD mixture also reduced the wash-out of polar petroleum metabolites (oxidized hydrocarbons) and the phytotoxicity of the lysimetric waters, especially in highly contaminated soils.

In another work, Fahid et al. (2019) [53] showed the effectiveness of bioaugmentation of hydrocarbon-degrading bacteria in diesel-polluted water with a plant presence. Floating treatment wetlands (FTWs) were planted with *Phragmites australis* and a hydrocarbon-degrading consortium (*Acinetobacter* sp. BRRH61, *Bacillus megaterium* RGR14, and *Acinetobacter iwoffii* AKR1) was inoculated. These combined conditions not only increased hydrocarbon degradation but also reduced toxicity in plant biomass.

Hydrocarbon bioremediation can also be stimulated by vermiremediation if soil pores have a low diameter (lower than 20 nm), which can hamper bacterial attack and hydrocarbon biodegradation. In this case, earthworms play a fundamental role in widening soil pores and making PAHs bioavailable. Furthermore, by both passive diffusion through the dermis and ingestion, earthworms absorb PAHs, making them less hazardous. Finally, they improve soil quality through the excretion of nutrients, which stimulate bacterial proliferation, increasing their biomass and enzymatic activities [54].

Koolivand et al. [55] investigated the biodegradation of PAHs using a bacterial consortium degrading hydrocarbons (BC), vermiremediation with *Eisenia fetida* (VC), and a combination of both (BCVC) in a kerosene solution. The bacterial consortium (composed of *Acinetobacter radioresistens* strain KA2 and *Enterobacter hormaechei* strain KA3) was isolated from the petroleum oil sludge of an oil refinery plant located in Iran. *Eisenia fetida* was used for evaluating the toxicity of the hydrocarbons (survival and weight loss of adults and number of cocoons). The kerosene concentration varied between 5, 10, and 20 g/kg. The results showed that at 12 weeks, the removal percentages were 81–83% in BC, 31–49% in VC, and 85–91% in BCVC, respectively, suggesting the synergistic effect of bacteria and worms in PAH bioremediation.

Nanomaterials have also been used to support microbial communities in the bioremediation of contaminants, including PAHs. In fact, nanomaterials (such as iron nanoparticles) can be used as co-promoters, which can immobilize contaminants [56,57]. For example, Kumari et al. [57] reported the successful use of iron nanoparticles for stimulating biosurfactant production in *Nocardiopsis* MSA13A, since Fe is necessary for its

synthesis [57]. In another work, Parthipan et al. [56] used a combination of a biosurfactant (5 mg/L), (produced by *Bacillus subtilis* A1) and iron nanoparticles (10 mg/L) for promoting PAH biodegradation by a bacterial consortium. The latter comprised *Pseudomonas stutzeri* NA3 and *Acinetobacter baumannii* MN3. The degradation of mixed PAHs (anthracene, pyrene, and benzo(a)pyrene) was enhanced by 52.6% (consortium alone), 65.7% (nanoparticles + consortium), 68% (biosurfactant + consortium) and 85% (nanoparticles + biosurfactant + consortium) of the initial concentration (500 mg/L) in 20 days. Adding the biosurfactant increased PAH bioavailability in the aqueous phase, and the nanoparticles stimulated bacterial biomass growth and PAH adsorption on their surfaces, assisting in the contact of bacterial cells with PAHs.

These laboratory and field studies have overall made it possible to define the main metabolic pathways for hydrocarbon biodegradation.

## 5. Metabolic Degradation Pathways

The bacterial catabolic metabolism for hydrocarbon degradation can be aerobic or anaerobic. The concentration of oxygen affects degradation rates; in fact, under aerobic conditions, biodegradation is faster and more efficient because oxygen is the final electron acceptor. Under anaerobic conditions, the final acceptor of electrons can be nitrate, sulfate, or an iron molecule, and biodegradation is slower and often negligible if compared to degradation under aerobic conditions [5,7,9].

In aerobic conditions, hydrocarbons are channeled into degradative pathways, which converge on the tricarboxylic acid cycle (TCA cycle or Krebs cycle) and lead to the complete oxidation of the substrate with the formation of $CO_2$ and NADH. The latter transfers electrons to the transport chain, which has $O_2$ as its final acceptor, and whose high redox potential (+0.82 mV) makes possible a high energy yield. Consequently, a discrete part of a substrate can be used for biosynthesis, permitting bacteria to multiply rapidly. As the number of individuals increases, substrate consumption also increases [8].

Hydrocarbon aerobic biodegradation occurs through numerous reactions, which can be divided into peripheral metabolic and central metabolic pathways.

Peripheral metabolic pathways convert a large proportion of hydrocarbons into a limited number of key intermediates.

The first stage of the aerobic degradation of hydrocarbons consists of the introduction of oxygen into the substrate which both makes the molecule more water soluble and increases its reactivity. Oxygen is introduced in the form of a hydroxyl group (-OH) and derives from molecular oxygen ($O_2$). The oxygenation reaction is catalyzed by enzymes called oxygenases.

There are two types of oxygenases: monooxygenases (Figure 2A) which introduce a single oxygen atom into the substrate, while the second one is reduced to water; and dioxygenases, which introduce both oxygen atoms into the substrate (Figure 2B).

Since oxygenation reactions require the utilization of reduced NADH or NADPH, these reactions can only occur within a bacterial cell, because these coenzymes rapidly degrade in the extracellular environment. This means that oxygenases can only act inside bacterial cells and that a hydrocarbon can undergo oxidation reactions only after being transported inside a cell. Monooxygenases can act on both aromatic and aliphatic hydrocarbons, while dioxygenases act mainly on aromatic hydrocarbons. In the latter case, dioxygenases involved in the activation of an aromatic molecule and dioxygenases involved in the opening of an aromatic ring can be distinguished. Both the activation and the opening of an aromatic ring occur through the introduction of both oxygen atoms into a substrate. The dioxygenases involved in ring opening do not require the intervention of reduced pyridine nucleotides [8].

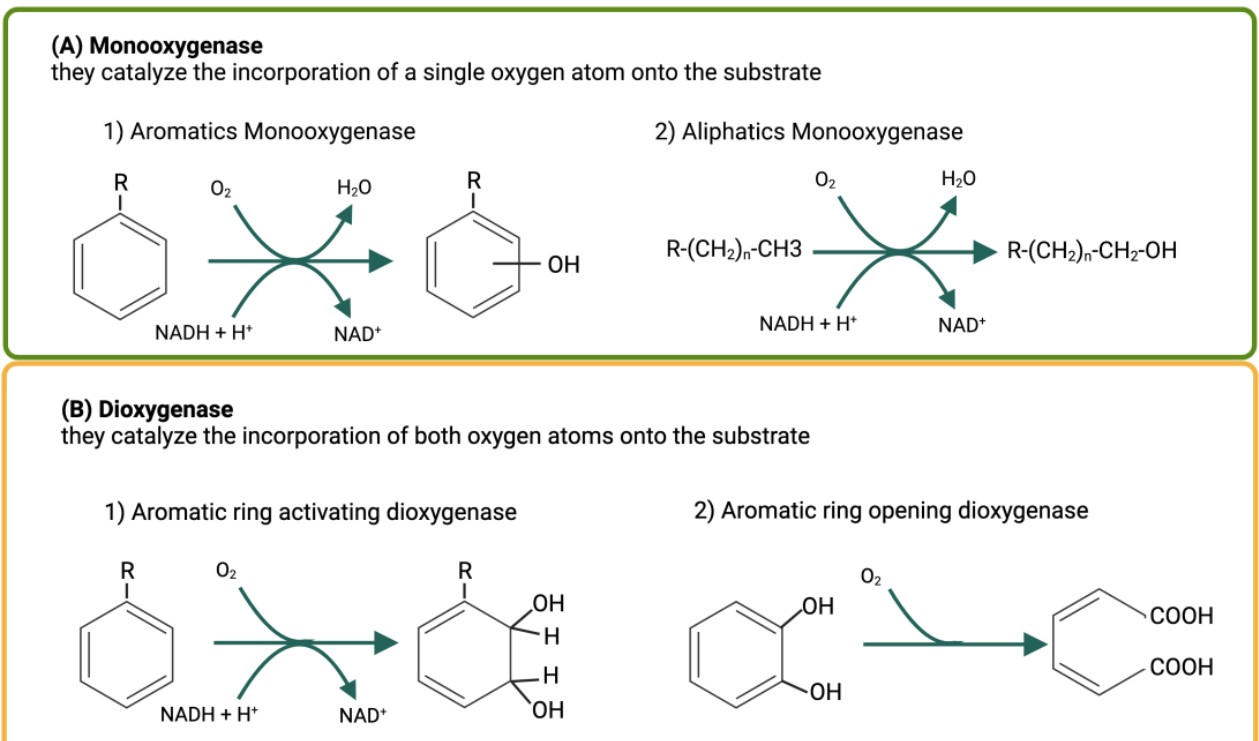

**Figure 2.** (**A**) Monooxygenases introduce one molecular oxygen atom into aromatic rings (on the left) or alkyl chains (on the right). (**B**) Dioxygenases introduce two oxygen atoms into aromatic substrates, activating an aromatic ring by forming dihydroxy compounds (on the left) or causing ring rupture of previously activated compounds (on the right).

The aerobic biodegradation of aliphatic hydrocarbons is reported in Figure 3. Some bacteria attack alkanes through an initial oxygenation of a sub-terminal methyl group; in this case, a secondary alcohol is formed, which is subsequently oxidized to a ketone and then to an ester. The hydrolysis of the ester bond leads to the formation of a fatty acid and an alcohol, which are subsequently oxidized to form the corresponding fatty acid (Figure 3A, on the left). Most bacteria degrade aliphatic hydrocarbons with the initial oxygenation of the terminal methyl group, which is in turn converted into an alcohol group. This is followed by a series of oxidations that lead to the conversion of alcohol into aldehyde groups and then into carboxyl groups (Figure 3B, on the right). In both pathways (sub-terminal and terminal oxidation) a fatty acid is obtained which can be metabolized by β-oxidation to produce acetyl-CoA which enters the Krebs cycle (also called tricarboxylic acid cycle -TCA- or citric acid cycle) [8].

Many aromatic hydrocarbons are also sources of carbon and energy for various aerobic organisms. Figure 4 reports naphthalene biodegradation. Starting from the initial substrate, an intermediate is formed consisting of a single aromatic ring carrying two hydroxyl groups (dihydroxylate), called catechol (ortho-diphenol). The ring cleavage is catalyzed by the enzyme catechol 1,2 dioxygenase in the ortho position and occurs through the introduction of two oxygen atoms onto the carbon atoms bearing the hydroxyl group. The bond between the carbon atoms in positions 1 and 2 is then broken, leading to the formation of cis, cis-muconic acid, which, through the formation of a lactone, is converted into succinate and acetyl-CoA (Figure 4, right side).

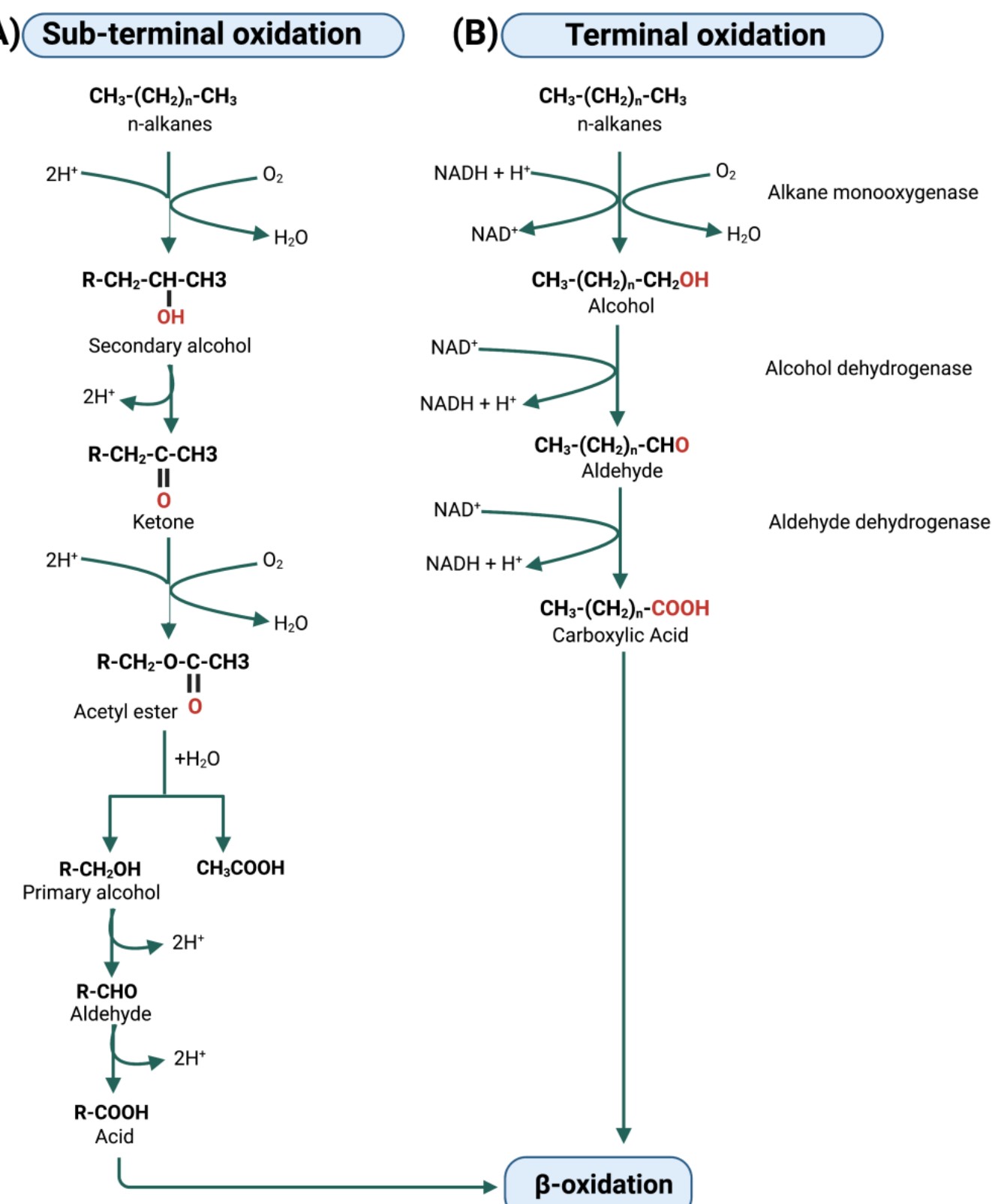

**Figure 3.** (**A**) Biodegradation of aliphatic hydrocarbons (n-alkanes) by an oxidation of the sub-terminal methyl group and (**B**) of the terminal methyl group.

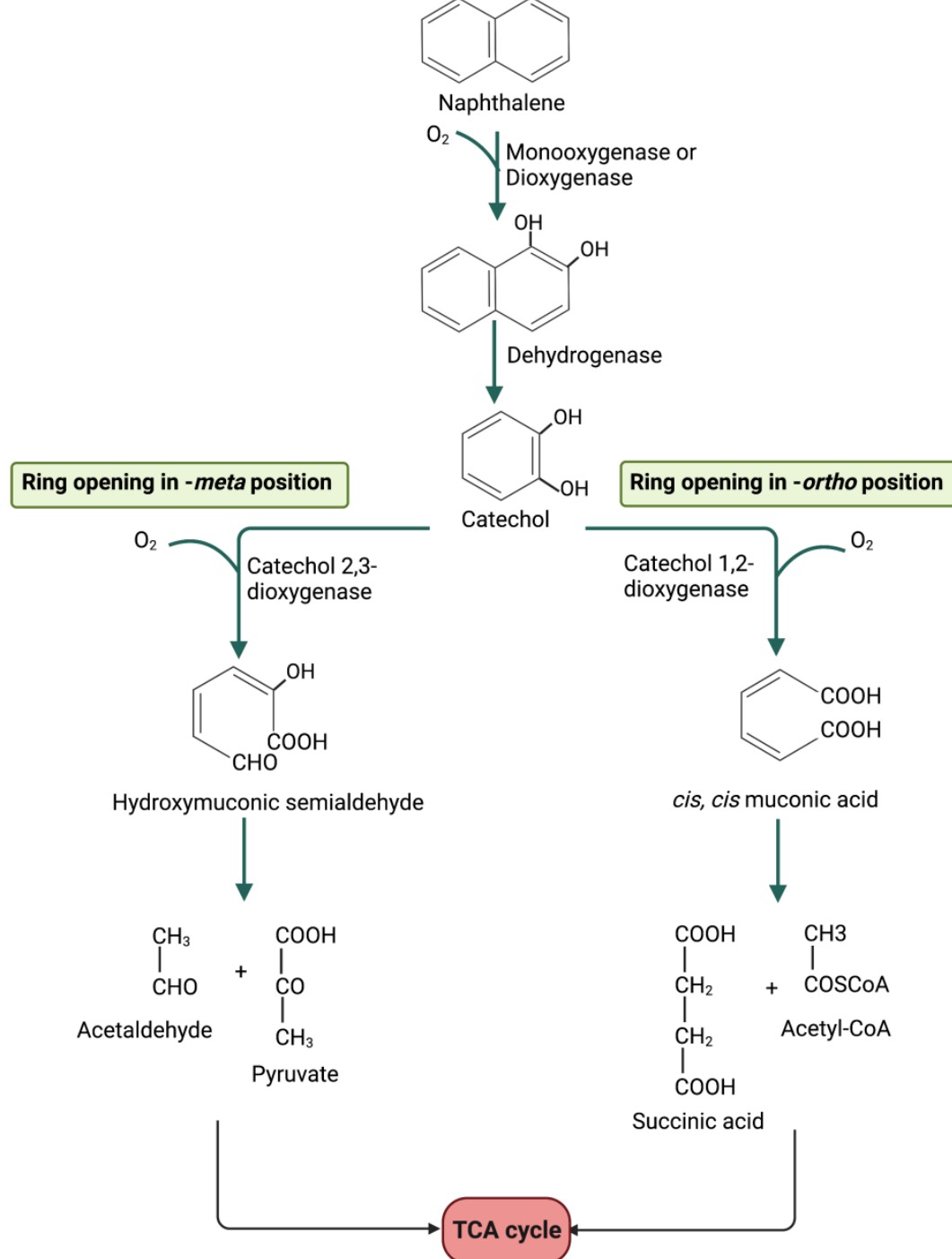

**Figure 4.** Biodegradation of aromatic hydrocarbons (e.g., naphthalene). The peripheral pathways lead to the formation of a dihydroxy intermediate (catechol) which is converted into linear intermediates. The linear molecules can be further processed to obtain compounds that enter the central metabolism of the bacterial cell (e.g., TCA cycle).

The ring cleavage in the meta position is catalyzed by the enzyme catechol 2,3 dioxygenase, which causes the opening of the ring between a hydroxylated carbon atom and the adjacent non-hydroxylated one. A hydroxymuconic semialdehyde is formed, which is transformed by subsequent reactions into pyruvate and acetaldehyde (Figure 4, left side).

Figure 5 summarizes the main degradative reactions of aliphatic and aromatic hydrocarbons, divided into peripheral (A), central (B) pathways, and (C) TCA cycle.

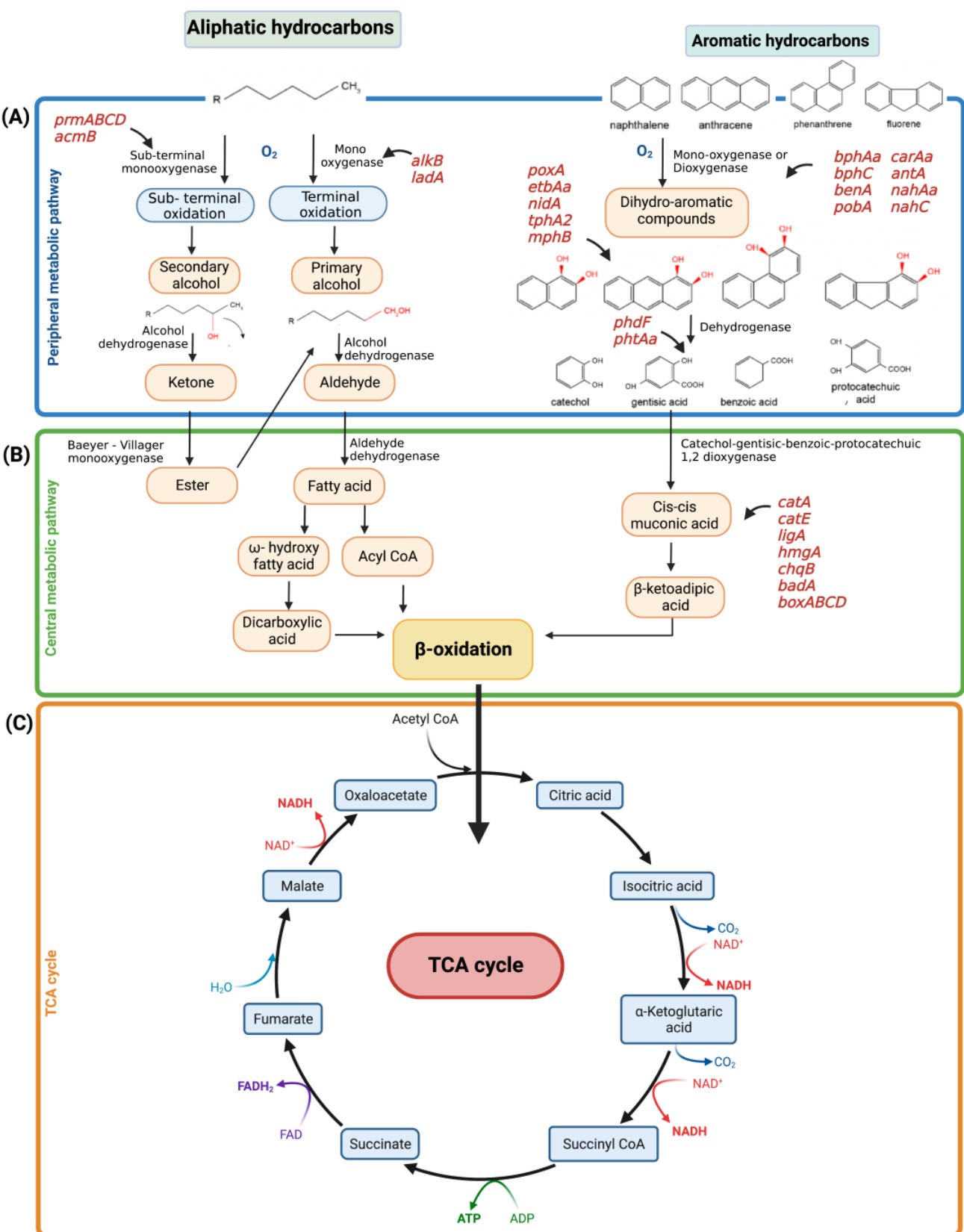

**Figure 5.** Main bacterial metabolic aerobic pathways of aliphatic and aromatic hydrocarbons. The pathways are subdivided into (**A**) peripheral metabolic pathway (blue rectangle), (**B**) central metabolic pathway (green rectangle); the genes encoding for enzymes involved in A and B metabolism are reported in red; (**C**) Krebs cycle: TCA cycle (orange rectangle).

The genes encoding enzymes involved in the peripheral metabolic pathways of aliphatic hydrocarbons are *alkB, ladA, prmABCD,* and *acmAB* (Figure 5A, left side of the blue rectangle).

The alkB gene codes for the enzyme alkane 1-monooxygenase, which is an integral membrane protein and catalyzes the initial hydroxylation of n-alkanes with a number of carbon atoms between 5 and 12 (C5–C12), oxidizing them to 1-alkanols. *AlkB* has been found in the *Pseudomonas* and *Rhodococcus* genera [4,58]. On the other hand, the *ladA* (long-chain alkane monooxygenase) gene participates in the terminal oxidation of n-alkanes by encoding a flavoprotein that inserts an oxygen molecule into long-chain alkanes. The *ladA* gene has been found in *Pseudomonas*, *Acinetobacter*, *Achromobacter,* and *Delftia* [59,60].

The genes encoding enzymes involved in the peripheral metabolic pathways of aliphatic hydrocarbons are *bphAa, bphC, benA, pobA, carAa, antA, nahAa, nahAb, nahC, nagG, poxA, etbAa, nidA, phdF, pht3, phtAa, tphA2, cmtAb, cmtC, hcaE,* and *mhpB* (Figure 5A, right side of the blue rectangle). These genes usually encode ring-hydroxylating dioxygenases and monooxygenases, which have the role of breaking down the aromatic ring. For example, biphenyl 2,3-dioxygenase, encoded by *bphAa*, is involved in the conversion of biphenyl to benzoate (Wang et al., 2021).

In particular, the *benA* and *pobA* genes encoding the benzoate 1,2 dioxygenase and hydrobenzoate 3-monooxygenase enzymes are fundamental for the first degradative phases of aromatic hydrocarbons [4].

In the central metabolic pathways, (Figure 5B) the intermediates produced by the peripheral metabolic pathways are further processed to obtain fatty acid molecules for their subsequent β-oxidation and final entrance into the tricarboxylic acid cycle (TCA) [61]. Hydrocarbon central pathways involve several genes for aromatics degradation such as *catAE, pcaG, ligA, hmgA, chqB, badA,* and the *boxABCD* cluster genes (Figure 5B, green rectangle, on the right). The latter encode for benzoyl-CoA 2,3 epoxidase, which transforms benzoate into its intermediates directed at the Krebs cycle [4] (Figure 5C). Once in the Krebs cycle (TCA cycle), the intermediates undergo oxidative phosphorylation.

The overall genes encoding the enzymes involved in the peripheral and central degradative metabolic pathways of aliphatic and aromatic hydrocarbons are summarized in Table 3.

The genes encoding enzymes involved in the degradation and metabolism of xenobiotic substances are generally located in plasmids [9], and this can assist in a prompt response to hydrocarbon contamination. However, the possibility that aerobic and anaerobic conditions occur at the same time, and that both high- and low-molecular-weight hydrocarbons are bioavailable in the same environment is very unlikely in natural ecosystems. Consequently, most hydrocarbons persist in natural environments. For this reason, biostimulation and bioaugmentation techniques need to be applied for hydrocarbon removal.

**Table 3.** List of the genes coding for the main enzymes involved in the peripheral and central metabolic pathways for the degradation of aliphatic and aromatic hydrocarbons.

|  | Gene | Enzyme | Metabolism |
|---|---|---|---|
| Aliphatics | *alkB* | Alkane 1-monooxygenase | Peripheral |
|  | *ladA* | Long chain alkane monooxygenase | Peripheral |
|  | *prmABCD* | Propane monooxygenase | Peripheral |
|  | *acmAB* | Bayer-Villiger monooxygenase and esterase | Peripheral |

**Table 3.** *Cont.*

| | Gene | Enzyme | Metabolism |
|---|---|---|---|
| Aromatics | *benA* | Benzoate 1,2 dioxygenase | Peripheral |
| | *pobA* | Hydrobenzoate 3-monooxygenase | Peripheral |
| | *bphAa* | Biphenyl 2,3-dioxygenase | Peripheral |
| | *bphC* | Dihydroxyphenyl 2,3-dioxygenase | Peripheral |
| | *carAa* | Carbazole 1,9-dioxygenase | Peripheral |
| | *antA* | Anthranilate 1,2-dioxygenase | Peripheral |
| | *nahAa* | Naphthalene1,2-dioxygenase (α-subunit) | Peripheral |
| | *nahAb* | Naphthalene1,2-dioxygenase (β-subunit) | Peripheral |
| | *nahAc* | Naphthalene1,2-dioxygenase (ferrodoxin) | Peripheral |
| | *nahC* | 1,2-dihydroxy naphthalene dioxygenase | Peripheral |
| | *nagG* | Salicylate 5-hydroxylase | Peripheral |
| | *poxA* | Phenol hydroxylase | Peripheral |
| | *etbAa* | Ethylbenzene dioxygenase | Peripheral |
| | *nidA* | Phenanthrene dioxygenase | Peripheral |
| | *tphA2* | Terephthalate 1,2-dioxygenase | Peripheral |
| | *phdF* | 3,4-dihydroxy-phenanthrene dioxygenase | Peripheral |
| | *pht3* | Phthalate 4,5-dioxygenase | Peripheral |
| | *phtAa* | Phthalate 3,4-dioxygenase | Peripheral |
| | *cmtAb* | p-coumate dioxygenase | Peripheral |
| | *cmtC* | 2,3-dihydroxy-p-coumate 3,4-dioxygenase | Peripheral |
| | *hcaE* | 3-phenylpropanoate/trans-cinnamate dioxygenase | Peripheral |
| | *mhpB* | 2,3-dihydroxy phenylpropionate 1,2-dioxygenase | Peripheral |
| | *catA* | Catechol 1,2-dioxygenase | Central |
| | *catE* | Catechol 2,3-dioxygenase | Central |
| | *pcaG* | Protocateucate 3,4-dioxygenase | Central |
| | *hmgA* | Homogensite 1,2-dioxygenase | Central |
| | *ligA* | Protocateucate 4,5-dioxygenase | Central |
| | *chqB* | Hydroxyquinol 1,2-dioxygenase | Central |
| | *badA* | Benzoate-CoA ligase | Central |
| | *boxA* | Benzoyl-CoA 2,3-epoxidase (α-subunit) | Central |
| | *boxB* | Benzoyl-CoA 2,3-epoxidase (β-subunit) | Central |
| | *boxC* | Benzoyl-CoA dihydrodiol lyase | Central |
| | *boxD* | 3,4-dehydroadipyl-CoA-semialdehyde dehydrogenase | Central |

## 6. Conclusions

Hydrocarbons are ubiquitous and complex contaminants, and their diffuse environmental presence causes concern for ecosystems and human health. Several hydrocarbons have been classified as priority and hazardous substances. Bacterial metabolism can potentially degrade them if their amounts are not so high as to be toxic to microorganisms and hamper their activity. Biodegradation depends not only on the presence of specific bacterial enzymes for degrading them but also on hydrocarbon bioaccessibility. Recently, investigations have focused on biostimulation and bioaugmentation strategies for improving hydrocarbon removal using biosurfactants, plants, nanoparticles, and earthworms in different combinations. Further research to improve knowledge of biodegradation processes at the field scale, to be applied to real-world studies, is desirable. A key aspect to take into consideration when moving from the lab to the real world is site-specific biotic and abiotic conditions, which make each ecosystem a unique one.

**Author Contributions:** Conceptualization, A.B.C., E.P., L.R.; resources, A.B.C.; data curation, A.B.C., E.P. and L.R.; A.B.C., E.P. and L.R writing—original draft preparation, A.B.C., E.P.; writing—review and editing, A.B.C. and E.P.; visualization, A.B.C., E.P.; supervision, A.B.C. project administration, A.B.C.. All authors have contributed to the manuscript and have read and agreed to the published version of the manuscript.

**Funding:** This research received no external funding.

**Institutional Review Board Statement:** Not applicable.

**Informed Consent Statement:** Not applicable.

**Data Availability Statement:** Not applicable.

**Conflicts of Interest:** The authors declare no conflict of interest.

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
