# Peer review of "Recent Advances in Bacterial Degradation of Hydrocarbons"

_water, doi:10.3390/w15020375_

Round 1
Reviewer 1 Report
The review manuscript is clear and concise. The topic seems to be very actual and should be of interest to the readers of the journal and a more general audience. However, some questions should be answered.
Authors should keep in mind the review paper style, namely, the available data should be adequately demonstrated and discussed (critics, evaluation, not just writing the literature results) as well.
The topic is huge, and the subsections just follow each other, consider writing some general conclusions at the end of each chapter.
The manuscript is too short. Authors should write some other examples. Write some lines (with examples) about the effect of hydrocarbons on ecosystem and health issues.
The manuscript should include chapters on issues related to the stability, accumulation, mobility, toxicity and fate of hydrocarbons (DNAPL). There are studies about risk assessment and fate regarding on hydrocarbons as well, these should be added to the article.
Authors mentioned a combination of other remediation techniques, e.g. phytoremediation. Authors should include a chapter on other methods that could be combined with biodegradation. For example application of nanomaterials or plants species.
The English is not completely satisfactory; it needs a thorough revision. Abbreviations should have been clarified properly at their first appearance.
Author Response
The review manuscript is clear and concise. The topic seems to be very actual and should be of interest to the readers of the journal and a more general audience. However, some questions should be answered.
Authors should keep in mind the review paper style, namely, the available data should be adequately demonstrated and discussed (critics, evaluation, not just writing the literature results) as well.
Thank you for your useful comments. The overall manuscript has been substantially revised, many parts re-written and new information added, included evaluations, comments and conclusion at the end of each chapter.
The topic is huge, and the subsections just follow each other, consider writing some general conclusions at the end of each chapter.
The manuscript is too short. Authors should write some other examples. Write some lines (with examples) about the effect of hydrocarbons on ecosystem and health issues.
Thank you for your comments. We have added new information on the effects of hydrocarbons on ecosystem and health issues (see lines 107-109; 114-172).
The manuscript should include chapters on issues related to the stability, accumulation, mobility, toxicity and fate of hydrocarbons (DNAPL). There are studies about risk assessment and fate regarding on hydrocarbons as well, these should be added to the article.
Thank you for your comments. You can find information about stability, accumulation, mobility, toxicity and fate of hydrocarbons in the text, (see lines 99-113; 148-158; 162-167).
Authors mentioned a combination of other remediation techniques, e.g. phytoremediation. Authors should include a chapter on other methods that could be combined with biodegradation. For example application of nanomaterials or plants species.
In accordance with your suggestions, we have added other remediation techniques to be combined with microorganisms (see lines 295-378).
The English is not completely satisfactory; it needs a thorough revision. Abbreviations should have been clarified properly at their first appearance.
The abbreviations have been clarified and the text has been thoroughly revised by an English mother tongue

Reviewer 2 Report
The review entitled, “Recent advances in Bacterial Degradation of Hydrocarbons” comprises different types of hydrocarbons, their environmental fate and toxic effects, degradation of hydrocarbons using microbes, metabolic pathways involved in degradation, involvement of microbial consortia in hydrocarbon biodegradation and bioremediation. I industrial era, this is a burning issue. This review is timely and as per title must represent the recent advances in studies aspect.
· However, The MS is descriptive, lacks flow and mechanistic approach.
· It requires substantial language editing and proof reading for typographic and grammatical errors.
· Abstract must be rewritten. It must be reorganized.
· A figure depicting process of hydrocarbon degradation by microbes will be useful for the readers to understand the mechanism easily.
· Avoid too many paragraphs. If possible merge them.
· Avoid using terms like- thanks to
· Check figure legends. Their meanings are not clear.
· A table consisting of studies on microbes assisted hydrocarbon degradation is expected.
· There is no synthesis of new information or extensive literature review.
· Revise the conclusion. It should be based on the synthesis of new information obtained from literature review.
· Add future prospects of the reviewed topic.
Author Response
Reviewer 2
Comments and Suggestions for Authors
The review entitled, “Recent advances in Bacterial Degradation of Hydrocarbons” comprises different types of hydrocarbons, their environmental fate and toxic effects, degradation of hydrocarbons using microbes, metabolic pathways involved in degradation, involvement of microbial consortia in hydrocarbon biodegradation and bioremediation. I industrial era, this is a burning issue. This review is timely and as per title must represent the recent advances in studies aspect.
- However, The MS is descriptive, lacks flow and mechanistic approach. Thank you for your comments. The manuscript has been substantially revised and new information added, included evaluations, comments and conclusion at the end of each paragraph. Moreover, three figures (Figure 2, 3, and 4) and two Tables (Table 2 and 3) have been added in order to explain in details the biodegradative pathways.
- It requires substantial language editing and proof reading for typographic and grammatical errors. Thank you for your useful suggestions. The overall text has been thoroughly revised by an English mother tongue
- Abstract must be rewritten. It must be reorganized. The overall text, including the abstract has been rewritten and the English style checked by an English mother tongue.
- A figure depicting process of hydrocarbon degradation by microbes will be useful for the readers to understand the mechanism easily. In accordance with your suggestions, Figures 2, 3, 4 and 5 depicts hydrocarbon biodegradation.
- Avoid too many paragraphs. If possible merge them. Yes. In accordance with your suggestion, some paragraphs have been merged (4 paragraph has been merged with 5).
- Avoid using terms like- thanks to. Ok, Done
- Check figure legends. Their meanings are not clear. Thank you for your useful suggestions. We have improved all figure legends
- A table consisting of studies on microbes assisted hydrocarbon degradation is expected. Thank you for the suggestion. Done, see Table 2
- There is no synthesis of new information or extensive literature review. We have improved this part and added comments at the end of each paragraph.
- Revise the conclusion. It should be based on the synthesis of new information obtained from literature review. We have re-written the overall conclusions.
- Add future prospects of the reviewed topic. We have added this in the conclusions.
Round 2
Reviewer 1 Report
The manuscript improved well, Authors answered all the question arised.
Reviewer 2 Report
Dear Team, Thanks for the mail. The MS can be accepted.